# Efficient Statistical Assessment of Neural Network Corruption Robustness

**Karim Tit**
Thales Land and Air Systems, BU IAS
Rennes, France
Univ. Rennes, Inria, CNRS, IRISA
Rennes, France
karim.tit@inria.fr

**Teddy Furon**
Univ. Rennes, Inria, CNRS, IRISA
Rennes, France
teddy.furon@inria.fr

**Mathias Rousset**
Univ. Rennes, Inria, CNRS, IRMAR
Rennes, France
mathias.rousset@inria.fr

## Abstract

We quantify the robustness of a trained network to input uncertainties with a stochastic simulation inspired by the field of Statistical Reliability Engineering. The robustness assessment is cast as a statistical hypothesis test: the network is deemed as locally robust if the estimated probability of failure is lower than a critical level. The procedure is based on an Importance Splitting simulation generating samples of rare events. We derive theoretical guarantees that are non-asymptotic w.r.t. sample size. Experiments tackling large scale networks outline the efficiency of our method making a low number of calls to the network function.

## 1   Introduction

Despite state-of-the-art performances on many Computer Vision and NLP tasks, Deep Neural Networks (DNNs) have been shown to be sensitive to both adversarial and random perturbations [Gilmer et al., 2019, Franceschi et al., 2018]. Concerns about their safety and reliability have come forth as their applications move to critical fields, such as the defense sector or self-driving vehicles.

**Certification**   A posteriori certification aims at verifying the correct behavior of a trained network $f : \mathbb{R}^n \to \mathbb{R}^m$. This expected property is usually defined locally (a.k.a. instance-wise property): the network performs correctly in the neighborhood $\mathcal{V}(\mathbf{x}_o) \subset \mathbb{R}^n$ of a particular input $\mathbf{x}_o \in \mathbb{R}^n$. Let us denote $\iota(\cdot|\mathbf{x}_o) : \mathbb{R}^n \to \{0, 1\}$ the function indicating a violation of the expected property. The network is locally correct if $\iota(\mathbf{x}|\mathbf{x}_o) = 0$ for any $\mathbf{x} \in \mathcal{V}(\mathbf{x}_o)$.

In classification, the property takes the name of *robustness* and reads as: the output of the network remains unchanged over the neighborhood $\mathcal{V}(\mathbf{x}_o)$. It certifies that the network is robust against inputs corrupted by uncertainties of limited support or adversarial perturbations of constrained distortion.

The certification mechanism has two desired features as defined in [Singh et al., 2018]:

- Soundness: it does not certify the network when the property does not hold.

- Completeness: it does certify the network whenever the property holds.

35th Conference on Neural Information Processing Systems (NeurIPS 2021).

**Corruption robustness assessment**    Adversarial robustness corresponds to a worst-case analysis whereas *corruption robustness* considers random perturbations of the inputs. The key ingredient is the introduction of a statistical model $\pi_0$ of epistemic uncertainties occurring along the acquisition chain of the input. For instance, Franceschi et al. [2018] take Gaussian and uniform distributions over the $l_p$ ball $\mathcal{B}_{p,\epsilon}(\mathbf{x}_o)$ of radius $\epsilon$ centered on $\mathbf{x}_o$.

This recent trend goes with a *quantitative* assessment gauging to what extent a given property holds or does not hold. For instance, Webb et al. [2019] estimate the probability $p$ that a property is violated under a given statistical model of the inputs. This approach makes no assumption about the network under scrutiny as it is used as a black box. This grants the scalability to tackle deep networks. The main difficulty lies in the efficiency, *i.e.* the computational power needed to estimate weak probabilities. Their lack of soundness stems from the inability to determine if the probability $p$ of violation is exactly zero or too small to be estimated.

Section 2 presents a brief overview of robustness assessment procedures outlining the assumptions made about the network and their limitations.

**This work**    presents a scalable and efficient procedure assessing corruption robustness under a large panel of statistical models. It provides completeness and theoretical guarantees on the lack of soundness.

## 2    Related work

This section reviews the state-of-the-art in certification and corruption robustness assessment.

**Assessment by example**    Distortion constrained adversarial attacks, like PGD, look for property violations, a.k.a. adversarial examples, inside the ball $\mathcal{B}_{p,\epsilon}(\mathbf{x}_o)$. In the context of robustness, they take advantage of the fast computation of the gradient of the network function thanks to back-propagation. They are fast, complete but unsound a priori. The network is not certified if the attack succeeds, but a failure says nothing about the property: attacks are empirical processes without guarantees.

**Formal certification**    Using a SMT solver, [Katz et al., 2017] provide a sound and complete certification method, called ReLUplex, designed for Neural Networks with ReLU activation functions. However the same paper shows that the problem of sound and complete certification of neural networks (even restricted to ReLU activations) is NP-complete. Scaling to large modern networks seems difficult. In addition, although these formal methods are complete in theory, in practice the procedure might give up, terminating undecided with a 'timeout' if the underlying solver is too slow.

**Incomplete certification**    To gain scalability, researchers have proposed sound but by-design incomplete verification methods resorting to abstract domains which are usually convex approximations of the input domain. Singh et al. [2019] obtain significant speeding up compared to ReLUplex and other complete certifiers. They introduce a verification benchmark called ERAN (see Sect. 5). Weng et al. [2018] design another incomplete certification based on lower and upper linear functional bounds of Multi-Layer Perceptrons (MLPs) with ReLU activation. It is generalized to MLPs with general activation functions in [Zhang et al., 2018] and to Convolutions Neural Networks (CNNs) in [Boopathy et al., 2019, Weng et al., 2019]. All these certifications methods rely on lower bounds of the minimum distance of adversarial examples. They are thus pessimistic in the sense that they can reject many valid properties because the lower bound is not always tight enough. In order to be 'more complete', Salman et al. [2019] solve the optimal convex relaxation (only for specific problems on MNIST and CIFAR10 datasets) with extensive computational resources. They conclude that the tightness of the lower bounds cannot be improved, suggesting that this approach has found its limit.

**Statistical assessment**    Corruption robustness assumes a statistical model $\pi_0$ of noisy inputs like Gaussian or uniform distributions over the $l_p$ ball $\mathcal{B}_{p,\epsilon}(\mathbf{x}_o)$. Franceschi et al. [2018] study the robustness of both linear and deep neural networks. They obtain precise bounds for linear classifiers which they extend to non-linear classifiers with 'locally approximately flat decision boundaries'. Webb et al. [2019] introduce a robustness metric (the lower, the more robust) is defined by:

$$p := \pi_0(\iota(\mathbf{X}|\mathbf{x}_o) = 1) = \int_{\mathbb{R}^n} \iota(\mathbf{x}|\mathbf{x}_o)\pi_0(d\mathbf{x}). \tag{1}$$

The quantitative assessment is in stark contrast with the adversarial robustness literature taking a worst-case analysis. A connection is established when $\pi_0$ is the uniform distribution over the ball $\mathcal{B}_{p,\epsilon}(\mathbf{x}_o)$: the volume of the set of adversarial examples equals $p \, \mathrm{vol}(\mathcal{B}_{p,\epsilon}(\mathbf{x}_o))$. Baluta et al. [2021] name probability $p$ the adversarial density.

The main difficulty lies in the estimation of this integral, in particular when the event $\{\iota(\mathbf{X}|\mathbf{x}_o) = 1\}$ is rare under distribution $\pi_0$. Baluta et al. [2021] use a crude Monte Carlo simulation. Webb et al. [2019] use Importance Splitting (a.k.a. multi-level splitting) with a rejuvenative mechanism based on the Metropolis-Hastings algorithm. These two last works make no assumption about the network as their procedure uses it as a black box. This grants scalability (in the sense that it tackles deep networks). The efficiency of the statistical test/estimation procedure is measured by the runtime or the number of calls to the black-box model.

The quantitative assessment falls back to certification by taking a final decision: the network is *deemed* reliable if the probability of violation is smaller than $p_c > 0$, a critical probability set by the user. It strikes the trade-off between the lack of soundness and the efficiency: a low $p_c$ increases the soundness but requires estimating probabilities as low as $p_c$ which is computationally demanding.

# 3 Our approach to corruption robustness assessment

Our approach uses statistical hypothesis testing as a certification *surrogate*. As in [Baluta et al., 2021], the user sets a low critical probability $p_c$ and the test assesses whether $p$ is lower or larger. However, rather than a testing approach powered by crude Monte Carlo simulations, our workhorse is a more efficient Sequential Monte Carlo algorithm [Naesseth et al., 2019]. This so-called 'Last Particle' simulation was invented by Guyader et al. [2011] and is a variant of the Adapative Multi-Level Sampling employed by Webb et al. [2019]. We show that, with a carefully chosen termination condition, it is advantageous both in terms of computational efficiency and theoretical guarantees.

Sect. 3.1 presents the 'Last particle' simulation that Sect. 3.2 applies to statistical hypothesis testing in the framework of robustness assessment. Alg. 1 gives the pseudo-code of our procedure.

## 3.1 The Last Particle simulation

The goal of the Last Particle simulation is to efficiently generate samples drawn according to a reference probability distribution $\pi_0$ but in a region $\mathcal{R} := \{\mathbf{y} : h(\mathbf{y}) > 0\} \subset \mathbb{R}^n$ where $h : \mathbb{R}^n \to \mathbb{R}$, is the so-called the *score function*. Efficiency is the ability to perform this task using few calls to the score function, even when probability $\pi_0(\mathcal{R})$ is small.

The simulation manages a set of $N$ particles (*i.e.* samples) which are initially i.i.d. with respect to $\pi_0$. The name 'Last Particle' comes from the fact that the simulation 'kills' the sample with the lowest score at each step. The score of this last particle becomes the intermediate level $L_k$ at iteration $k$ (Alg. 1, line 6). Then, that particle is refreshed by sampling according to $\pi_0$ but conditioned on the event $\{h(\mathbf{X}) > L_k\}$. This sampling procedure is performed by $\mathsf{Gen}(L_k, 1)$ in line 11 and is detailed in Sect. 4. $\mathsf{Gen}(-\infty, N)$ then simply means sampling $N$ random vectors according to $\pi_0$ (line 3).

The algorithm stops when the number of iterations reaches integer $m$ or at any iteration $k$ if the intermediate threshold $L_k$ is positive which means the simulation has generated samples as required.

Consider the function $\Lambda : \mathbb{R} \to \mathbb{R}_+$ defined as

$$\Lambda(\ell) := -\log \pi_0(h(\mathbf{X}) > \ell). \tag{2}$$

This function is unknown in practice, but one can easily see that it is non decreasing.

During one run of Alg. 1, the intermediate levels are random variables following an increasing order by construction: $L_1 < L_2 < \cdots < L_k$. We here copy the main result of the Last Particle simulation:

**Theorem 1** ([Guyader et al., 2011])**.** *The variables* $\Lambda(L_1), \Lambda(L_2), \cdots$ *are distributed as the successive arrival times of a Poisson process with rate* $N$*:* $\Lambda(L_k) = 1/N \sum_{j=1}^{k} E_j$*, where* $E_j \overset{i.i.d.}{\sim} \mathcal{E}(1)$*.*

As the sum of i.i.d. exponential random variables is distributed[1] as a Gamma random variable, this theorem states that $\Lambda(L_k) \sim \Gamma(k, N)$ (*i.e.* scale $k$ and rate $N$).

---

[1]Here and after, $\sim$ denotes distributional equality between random variables.

**Algorithm 1** Robustness assessment with Last Particle simulation

---

**Require:** Number of particles $N$, critical probability level $p_c$, confidence interval level $\alpha$
**Ensure:** Cert
  1: Initialize: $p \leftarrow 1 - {}^1\!/_N, \quad k \leftarrow 1, \quad$ Cert $\leftarrow False, \quad$ Stop $\leftarrow False$
  2: $m \leftarrow$ Comp_m$(p_c, \alpha, N)$                                       ▷ See Sect. 3.3
  3: $\{\mathbf{x}_i\}_{i=1}^N \leftarrow$ Gen$(-\infty, N)$                                ▷ See Sect. 4
  4: **while** $k \leq m$ & Stop $= False$ **do**
  5:       $i^\star \leftarrow \arg\min_{i \in 1:N} h(\mathbf{x}_i)$
  6:       $L_k \leftarrow h(\mathbf{x}_{i^\star})$
  7:       **if** $L_k > 0$ **then**
  8:           Stop $\leftarrow True$
  9:           $P_{est} \leftarrow p^{k-1}$
10:       **end if**
11:       $\mathbf{x}_{i^\star} \leftarrow$ Gen$(L_k, 1)$                                  ▷ See Sect. 4
12:       $k \leftarrow k + 1$
13: **end while**
14: **if** Stop $= False$ **then**
15:       Cert $\leftarrow True$
16:       $P_{est} \leftarrow p_c$
17: **end if**
18: **return** Cert, $P_{est}$

---

## 3.2 Corruption robustness assessment as a statistical test

In the framework of robustness assessment of classifiers, the score function is related to the usual loss in the adversarial example literature:

$$h(\mathbf{x}) := \max_{k \neq c(\mathbf{x}_o)} f_k(\mathbf{x}) - f_{c(\mathbf{x}_o)}(\mathbf{x}), \tag{3}$$

where $f(\mathbf{x})$ represents the predicted probabilities (or logits) vector and $c(\mathbf{x}) := \arg\max_k f_k(\mathbf{x})$ is the predicted class for input $\mathbf{x}$. Note that $h(\mathbf{x}_o) < 0$ and that the violation indicator function of Sect. 1 is simply $\iota(\mathbf{x}|\mathbf{x}_o) = \mathbb{1}(h(\mathbf{x}) > 0)$. The input $\pi_0$ models the corruption distribution around $\mathbf{x}_o$. The probability of robustness violation defined is (1) writes as $p := \pi_0(h(\mathbf{X}) > 0)$.

Our approach establishes a hypothesis test parametrized by a low probability $p_c$ given by the user.

- $\mathcal{H}_0$: The probability of robustness violation $p > p_c$. The network should not be certified.
- $\mathcal{H}_1$: The probability of robustness violation $p < p_c$. The network can be certified.

For a given true probability of violation $p$, we establish the following properties.

**Proposition 1.** *The probability of false positive $P_{\mathsf{fp}}(p)$ equals:*

$$P_{\mathsf{fp}}(p) := \mathbb{P}(\mathsf{Cert} = \mathsf{True} | p > p_c) = \frac{\int_0^{-N \log p} t^m \mathrm{e}^{-t} dt}{\int_0^{+\infty} t^m \mathrm{e}^{-t} dt} = \frac{\gamma(m, -N \log p)}{\Gamma(m)}. \tag{4}$$

*Proof.* Certification means that, according the Alg. 1, even after $m$ loops, the intermediate threshold $L_m$ is still lower than $0$. This happens with probability:

$$P_{\mathsf{fp}}(p) = \mathbb{P}(L_m < 0) = \mathbb{P}(\Lambda(L_m) < \Lambda(0)) = \frac{\gamma(m, -N \log p)}{\Gamma(m)}, \tag{5}$$

since $\Lambda(L_m) \sim \Gamma(m, N)$ and $\Lambda(0) = -\log p$ ; $\gamma(s, x)$ being the lower incomplete gamma function.
$\square$

**Proposition 2.** *The probability of false negative $P_{\mathsf{fn}}(p)$ equals:*

$$P_{\mathsf{fn}}(p) := \mathbb{P}(\mathsf{Cert} = \mathsf{False} | p < p_c) = \frac{\int_{-N \log p}^{+\infty} t^m \mathrm{e}^{-t} dt}{\int_0^{+\infty} t^m \mathrm{e}^{-t} dt} = \frac{\overline{\gamma}(m, -N \log p)}{\Gamma(m)}. \tag{6}$$

*Proof.* The certification failed because $L_K > 0$ for some $K \leq m$, or equivalently $\Lambda(L_K) > \Lambda(0)$. Note that this was not true at iteration $K - 1$ (otherwise the while loop would have be broken earlier). In other words, $K - 1 = \sup\{i : \sum_{j=1}^{i} E_j < -N \log p, E_j \overset{\text{i.i.d.}}{\sim} \mathcal{E}(1)\}$, so that $K - 1$ follows the Poisson distribution $\mathcal{P}(-N \log p)$. The probability of false negative is the c.d.f. of $K - 1$ at $m - 1$:

$$P_{\mathsf{fn}}(p) = \mathbb{P}(K \leq m) = \mathbb{P}(K - 1 \leq m - 1) = \frac{\overline{\gamma}(m, -N \log p)}{\Gamma(m)}, \tag{7}$$

where $\overline{\gamma}(s, x)$ is the *upper* incomplete gamma function. $\qquad\square$

This shows that $P_{\mathsf{fn}}(p)$ is an increasing function and the worst case happens when $p$ converges to $p_c$:

$$\forall p < p_c, \; P_{\mathsf{fn}}(p) \leq P_{\mathsf{fn}}(p_c) = 1 - P_{\mathsf{fp}}(p_c). \tag{8}$$

Remark that the trade-off between false positive and false negative probabilities is hard at $p = p_c$. Yet, Eq. (7) tells that $P_{\mathsf{fn}}(p)$ is quickly vanishing as $p \to 0$, especially when $N$ is large.

### 3.3 Corruption robustness assessment as a certification problem

In the context of certification, we show that that i) our procedure is complete but not sound, ii) false positive probability drives the lack of soundness.

A false negative is not a bad event since it avoids us to certify when the probability $p$ of violation is not zero. At the same time, our procedure always certifies whenever the property holds since $P_{\mathsf{fn}}(0) = 0$. On the contrary, a false positive remains an error since we certify when $p > p_c > 0$. Let us quantify the lack of soundness by

$$P_{\mathsf{ns}}(p) := \mathbb{P}(\mathsf{Not\,Sound}\,|p) = \begin{cases} 1 - P_{\mathsf{fn}}(p) & \text{if } p < p_c \\ P_{\mathsf{fp}}(p) & \text{otherwise} \end{cases} \tag{9}$$

Let us recall that in our case it holds simply $1 - P_{\mathsf{fn}}(p) = P_{\mathsf{fp}}(p)$.

**Proposition 3.** *A suitable choice of the maximum number of iterations $m$ in Alg. 1 can control the lack of soundness by the critical probability $p_c$ and a required significance level $\alpha \in (0, 1)$ s.t.*

$$P_{\mathsf{ns}}(p) \leq \alpha, \forall p \geq p_c. \tag{10}$$

*Proof.* This amounts to enforce that $P_{\mathsf{fp}}(p) \leq \alpha, \forall p > p_c$. Since $-\log p$ is a decreasing function, the worst case occurs in (5) when $p \to p_c$. It is thus safe to ensure $P_{\mathsf{fp}}(p_c) = \alpha$. This is done by carefully selecting $m$ s.t. the $\alpha$-quantile of the r.v. $\Gamma(m, N)$ equals $-\log p_c$. The routine Comp_m in Alg. 1 solves this numerically with a line search (see Appendix A for some approximations). $\quad\square$

If we assume a Bayesian approach where the p.d.f. of $p$ is denoted by $f_P : [0, 1] \to \mathbb{R}_+$, then the probability of not being sound is given by

$$\mathbb{P}(\mathsf{Not\,Sound}) \;\; = \;\; \int_{0+}^{p_c} (1 - P_{\mathsf{fn}}(p)) f_P(p) dp + \int_{p_c}^{1} P_{\mathsf{fp}}(p) f_P(p) dp \tag{11}$$

$$\leq \;\; \int_{0+}^{p_c} f_P(p) dp + \alpha \int_{p_c}^{1} f_P(p) dp = \alpha + (1 - \alpha) \mathbb{P}(p < p_c). \tag{12}$$

The lack of soundness decreases if both $\alpha$ and $p_c$ are small. This makes the point with the state-of-the-art. Baluta et al. [2021] are unable to set $p_c$ to a low value because their simulation is based on a crude Monte Carlo, whereas Webb et al. [2019] do not give any guarantee similar to our level $\alpha$.

**Efficiency** Appendix A proposes approximated closed forms outlining that $m$ scales as $\log 1/p_c$. This is also visible in the typical values given in Table 1. A lower significance level moderately increases the number of iterations. Section 4 details how to sample a new particle at each iteration as needed in line 11, Alg. 1. This method consumes a fixed number of calls to the network. In total, the maximum number of calls scales as $O(\log 1/p_c)$. This is in stark contrast with [Baluta et al., 2021] where the number of calls is proportional to $1/p_c$. Note that this is a maximum: our procedure makes an early stop whenever $L_k > 0$ (line 8, Alg. 1) and outputs $Cert = False$ as well as failure probability estimate $P_{est}$.

Table 1: Maximum number of iterations $m$ and its approximation $\tilde{m}_1$ (see App. A)

| $N$ | $p_c$ | $\alpha = 0.1$ | | $\alpha = 0.01$ | | $\alpha = 0.001$ | |
|---|---|---|---|---|---|---|---|
| | | $m$ | $\tilde{m}_1$ | $m$ | $\tilde{m}_1$ | $m$ | $\tilde{m}_1$ |
| 20 | $10^{-10}$ | 489 | 489 | 512 | 514 | 529 | 532 |
| 20 | $10^{-30}$ | 1430 | 1431 | 1470 | 1471 | 1499 | 1502 |
| 10 | $10^{-10}$ | 251 | 251 | 267 | 269 | 280 | 283 |
| 10 | $10^{-30}$ | 726 | 726 | 754 | 755 | 774 | 777 |
| 2 | $10^{-10}$ | 56 | 56 | 64 | 65 | 69 | 73 |
| 2 | $10^{-30}$ | 154 | 155 | 167 | 169 | 177 | 180 |

## 4 Sampling procedures

This section details the crucial ingredient of our procedure: sampling a new input $\mathbf{X}$ whose score $h(\mathbf{X})$ is above a given level $L$. This random generator is called $\mathrm{Gen}(L, 1)$ in line 11, Alg. 1. Appendix B considers a case where the statistical model $\pi_0$ and the network are so simple that this sampling is easy. This section details more general scenarios making no assumption about the score function. Our sampling is a rejection procedure relying on *reversible proposals* and *transformations*.

### 4.1 Reversible proposals

We call a (parametric) proposal any a random function $K : \mathbb{R}^n \times \mathbb{R}_+ \to \mathbb{R}^n$. Iterations of i.i.d. proposal generate a Markov chain which is said to be *reversible (detailed balance)* with respect to the distribution $\pi_0$ if the following assertion holds:

$$\forall s > 0, \ \mathbf{X} \sim \pi_0 \Rightarrow (\mathbf{X}, K(\mathbf{X}, s)) \sim (K(\mathbf{X}, s), \mathbf{X}). \tag{13}$$

A simple example for $\pi_0 = \mathcal{N}(\mathbf{0}_n; \sigma^2 \mathbf{I}_n)$ is given in [Guyader et al., 2011]:

$$K(\mathbf{X}, s) := \frac{\mathbf{X} + s\mathbf{N}}{\sqrt{1 + s^2}} \quad \text{with} \ \mathbf{N} \sim \mathcal{N}(\mathbf{0}_n; \sigma^2 \mathbf{I}_n). \tag{14}$$

The rejection method described in Alg. 2 takes as input a set $\mathcal{X}$ of particles whose score is larger than $L$. It randomly picks one particle in $\mathcal{X}$ and applies $t$ times a fresh proposal, followed by a rejection based on the score. If the selected sample is a realization of the distribution $\pi_0$ conditioned by a score larger than $L$, then one application of the proposal keeps $\pi_0$ invariant while the rejection ensures that the score remains above $L$. By induction, iterating maintains these two properties, and in fact leaves invariant the conditional distribution thanks to reversibility (see [Guyader et al., 2011] and App. E).

Alg. 1 uses the procedure of Alg. 2 as follows. At iteration $k$, $L$ is indeed $L_k$, *i.e.* the score of the 'last' particle $\mathbf{x}_{i^\star}$, and $\mathcal{X} = \{\mathbf{x}_i\}_{i \neq i^\star}$ which contains $(N-1)$ particles whose score is larger than $L$. The output is one 'fresh' particle and the number of particles equals $N$ from one iteration to another.

The parameter $s$ plays the role of *strength*: $s = 0$ implies that the proposal just copies the input, while $s \to +\infty$ means that $K(\mathbf{x}, s)$ does not depend on $\mathbf{x}$. The proposal strength $s$ is thus important. With a small value, the proposal makes small moves. A large value explores faster but leads to higher rejection rate. Appendix D presents a strategy to automatically control its value depending on the past behavior of the algorithm in order to maintain a given rejection rate.

Theoretically, under some irreducibility assumption, an infinity of iterations in Alg. 2 provides a fresh particle statistically independent of the particles in $\mathcal{X}$ as needed in Alg. 1:

**Proposition 4.** *Assume that, the proposal $K(\mathbf{x}, s)$ has a density bounded from below uniformly in $\mathbf{x}$ and $s \geq s_0$. Then the distribution of $\Lambda(L_m)$ converges towards the Gamma distribution $\Gamma(N, m)$ exponentially fast with the number $t$ of proposal applications.*

*Proof and Remarks.* The proof is given in Appendix E and uses a classical probabilistic coupling argument. It requires the lower bound assumption which is a form of strong irreducibility of the proposal. This is compliant with the proposals used in this work. In particular all the formulas given in Prop. 1 and after hold true asymptotically for large $t$. $\square$

**Algorithm 2** Sampling one particle $\mathsf{Gen}(L, 1)$

---

**Require:** threshold $L$, finite set $\mathcal{X}$ of particles whose score is larger than $L$
**Ensure:** new particle $\mathbf{X}$
 1: $\mathbf{X} \leftarrow \mathcal{U}(\mathcal{X})$        ▷ Draw uniformly a particle in $\mathcal{X}$
 2: **for** $k = 1 : t$ **do**
 3:     $\mathbf{Z} \leftarrow K(\mathbf{X}, s)$        ▷ $\pi_0$ reversible proposal. See Sect. 4.1
 4:     **if** $h(\mathbf{Z}) > L$ **then**        ▷ Rejection
 5:        $\mathbf{X} \leftarrow \mathbf{Z}$
 6:     **end if**
 7: **end for**
 8: **return** $\mathbf{X}$

---

In practice, we choose the number $t$ of iterations approximately proportional to the inverse of rejection rate, maintained approximately constant by tuning the proposal strength $s$ (see App. D).

Refreshing a particle consumes $t$ calls to the score function. This is done *once* per iteration of Alg. 1. Therefore, our method globally consumes $O(t \log 1/p_c)$ calls. This means that the figures in Table 1 are to be multiplied by $t$. Webb et al. [2019] also manage a sample of size $N$, but *all* the particles are separately refreshed at *each* iteration by applying $t$ Metropolis-Hasting transitions. Their number of calls per iteration is $N$ times larger than our. Moreover, their typical setup is $N \approx 1000$ and $t \approx 1000$, while ours is $N \approx 2$ and $t \approx 50$. Our complexity is thus smaller by 4 orders of magnitude.

### 4.2 Isoprobabilistic transformation

The proposal (14) is simple but reversible only w.r.t. the normal distribution. The transformation method is well known in the field of Statistical Reliability Engineering [Melchers and Beck, 2018]. It amounts to work with a latent random vector $\mathbf{G} \sim \mathcal{N}(\mathbf{0}_d; \mathbf{I}_d)$ and to apply the transformation $\mathbf{X} = T(\mathbf{G}, \mathbf{x}_o)$ mapping the normal distribution to the reference model $\pi_0$. Some well known examples are:

- $\mathbf{X} \sim \mathcal{N}(\mathbf{x}_o, \sigma^2 \mathbf{I}_n)$: $d = n$ and $T(\mathbf{G}, \mathbf{x}_o) = \mathbf{x}_o + \sigma \mathbf{G}$
- $\mathbf{X} \sim \mathcal{U}(\mathcal{B}_{+\infty, \epsilon}(\mathbf{x}_o))$: $d = n$ and $T(\mathbf{G}, \mathbf{x}_o) = \mathbf{x}_o + \epsilon(2\Phi^{-1}(\mathbf{G}) - 1)$ (component-wise)
- $\mathbf{X} \sim \mathcal{U}(\mathcal{B}_{2, \epsilon}(\mathbf{x}_o))$: $d = n + 2$ and $T(\mathbf{G}, \mathbf{x}_o) = \mathbf{x}_o + \epsilon \mathbf{G}(1 : n)/\|\mathbf{G}\|_2$

More complex examples are inverse Rosenblatt or Nataf transformations [Melchers and Beck, 2018].

This transformation is composed with $h$ to redefine the score function $h_G = h \circ T$ that applies on latent vector $\mathbf{G}$, *i.e.* random vectors suitable for the proposal (14). This amounts to use Alg. 1 directly on the latent variable with score function $h_G$ and in conjunction with Alg. 2 and proposal (14).

## 5 Experimental evaluation

This section presents experimental results on ACAS Xu, MNIST, and ImageNet datasets with some trained classifications networks listed in App. F.3 together with implementation details. Experiences were run on a laptop PC (CPU=Intel(R) Core(TM) i7-9750H, GPU=GeForce RTX 2070) except for experiences on ImageNet which were run on a Nvidia V100 GPU.

### 5.1 Idealized case

This section considers a setup where $\pi_0 = \mathcal{N}(\mathbf{x}_o; \sigma^2 \mathbf{I}_n)$ and score function $h$ is linear. This setup is ideal because sampling a fresh particle is straightforward (*i.e.* without Alg. 2) as shown in App. B.

Fig. 1 shows the impact of $N$. In terms of hypothesis testing (see Sect. 3.2), a larger $N$ yields steeper functions: $P_{\mathsf{fp}}(p)$ (resp. $P_{\mathsf{fn}}(p)$) quickly vanishes to zero as $p$ gets larger (resp. smaller) than $p_c$. In terms of certification (see Sect. 3.3), a small $N$ is not a bad choice: the probability $P_{\mathsf{ns}}(p)$ of not being sound takes lower values in the range $p < p_c$. For $p > p_c$, $P_{\mathsf{ns}}(p)$ is lower than $\alpha$ (as stated by Prop. 3) but converges to 0 more slowly. Last but not least, the procedure makes only 167 calls to the score function for $N = 2$, instead of 1470 for $N = 20$.

Table 2: ACAS Xu – Confusion matrix comparing ERAN [DeepPoly+MILP] and Last Particle $[N = 2, p_c = 10^{-50}, t = 40]$

|  |  | ERAN | | | |
| --- | --- | --- | --- | --- | --- |
|  |  | Certified | Uncertified | Infeasible | TimeOut |
| Last Particle | Certified | 107 | 9 | 1 | 1 |
|  | Uncertified | 0 | 103 | 4 | 0 |

## 5.2 ACAS Xu

We evaluate our method on the ACAS Xu (Airborne Collision Avoidance System X for unmanned aircrafts) case study [Owen et al., 2019]. It consists in 45 neural networks used to approximately compress a large lookup table (2GB) containing discrete decisions ('Clear-of-conflicts', 'weak right', 'strong right', 'weak left', or 'strong left') as well as 5 input/output properties. This makes $45 * 5 = 225$ cases. We compare our method with the complete certification based on DeepPoly [Singh et al., 2019] and Mixed-Integer Programming from the ERAN benchmark.

Table 2 contains the confusion matrix taking into account the cases for which the ERAN complete certification fails because the Gurobi optimizer either outputs an 'infeasible' status or reaches a timeout (set to 600 seconds). Unsurprisingly, our method is complete in the sense that it certifies all cases certified by ERAN. It is not sound as it admits 9 false positives. This is due to the critical probability $p_c$ which is not low enough (the decisions were exactly the same over 10 runs). Yet, our method takes a decision on the 6 unsolved cases by ERAN. In addition our method is faster for all ACAS Xu properties except for the property 4, confer figure 2.

## 5.3 MNIST

We compare our procedure with with the DeepPoly *incomplete* certification on MNIST [LeCun et al., 1990] with 4 neural networks from the ERAN benchmark (see App. F.3). We focus on $L_\infty$ uniform robustness since the implementation provided for DeepPoly cannot deal with $L_2$ norms. We run our algorithm with $N = 2, p_c = 10^{-35}$ and $t = 40$. As in ACAS Xu experiment, our method runs faster than the ERAN method as shown in table 3. Interestingly, the average runtime of our method decreases with larger $\epsilon$ since the probability $p$ of violation is bigger, whereas DeepPoly computation time increases with the size of the input space tested. On the one hand DeepPoly provides an efficient lower bound to both corruption and adversarial robustness, on the other hand our method provides a fast upper bound. 10 independent LP simulations (runs) on the same image always give the same output and the standard deviation is thus empirically negligible in our setting.

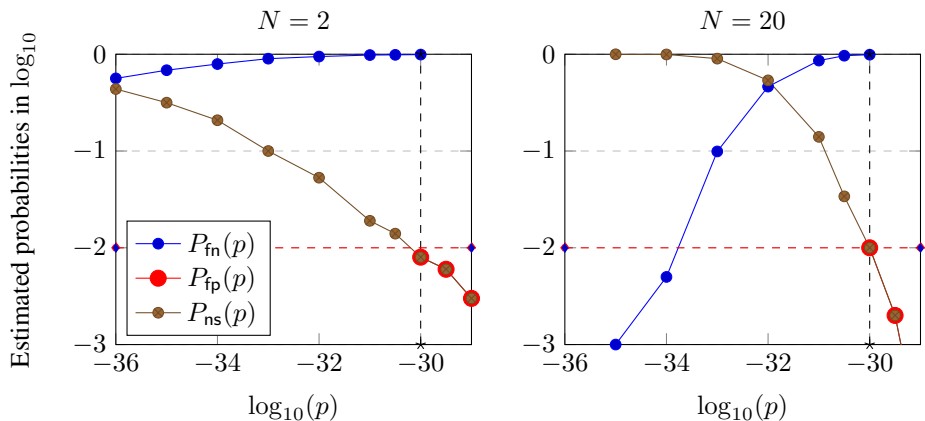

Figure 1: Estimated probabilities of false positive, false negative, and not sound certification, vs. true violation probability $p$ in the ideal setup where $p_c = 10^{-30}$, $\alpha = 0.01$. Estimation over 1000 runs.

Table 3: MNIST – Comparison ERAN [DeepPoly] and Last Particle $[N = 2, p_c = 10^{-35}, t = 40]$

| | ERAN | | Last Particle | |
|---|---|---|---|---|
| $\epsilon$ | Certified (%) | runtime (sec.) | Certified (%) | runtime (sec.) |
| 0.015 | 82 | 5.69 | 99 | $1.04 \pm 0.005$ |
| 0.03 | 62 | 5.92 | 97 | $1.03 \pm 0.01$ |
| 0.06 | 28 | 8.13 | 93 | $1.00 \pm 0.01$ |
| 0.1 | 22 | 8.84 | 85 | $0.96 \pm 0.02$ |

Table 4: ImageNet - Last Particle $[N = 2, p_c = 10^{-15}, t = 20]$

| Network | $\epsilon$ | Avg. runtime (in sec. $\pm std$) | Avg. number of calls | Certified (%) |
|---|---|---|---|---|
| MobileNet | 0.02 | $20.78 \pm 0.74$ | 1388 | 71 |
| | 0.03 | $18.74 \pm 0.18$ | 1274 | 64 |
| | 0.06 | $14.5 \pm 0.11$ | 1037 | 50 |
| ResNet50 | 0.02 | $33.86 \pm 1.14$ | 1537 | 81 |
| | 0.03 | $31.38 \pm 0.48$ | 1434 | 71 |
| | 0.06 | $25.51 \pm 0.67$ | 1160 | 59 |

## 5.4 ImageNet

For the last experiment, our method analyses 2 neural networks (ResNet50 et MobileNet) with 100 test images from ImageNet dataset [Deng et al., 2009] correctly classified by each network. These experiments were run on a Nvida V100 GPU. The average number of calls reported is rounded up and the average runtime is for a pass over one image. The robustness is again defined against noise uniformly distributed over $L_\infty$ of radius $\varepsilon$. As one can notice, the compute time increases reasonably the input space dimension and network size.

## 6   Conclusion

The paper proposes a statistical simulation to make assessment on corruption robustness. It looks at this problem from a hypothesis testing (false positive/ false negative) and from a certification (completeness / soundness) points of view. The procedure is scalable, efficient, complete and comes with guarantees on the lack of soundness. There are two limitations: 1) The Last Particle simulation is sequential, which is not GPU friendly. Yet, we provide a code processing several inputs $\mathbf{x}_o$ in parallel. 2) Our procedure is general as it uses the network as a black-box classifier. But, it does not exploit its gradient easily computed thanks to backpropagation. More sophisticated mixing kernels using gradients information (e.g. Langevin Monte Carlo, Hamiltonian MC) can accelerate convergence.

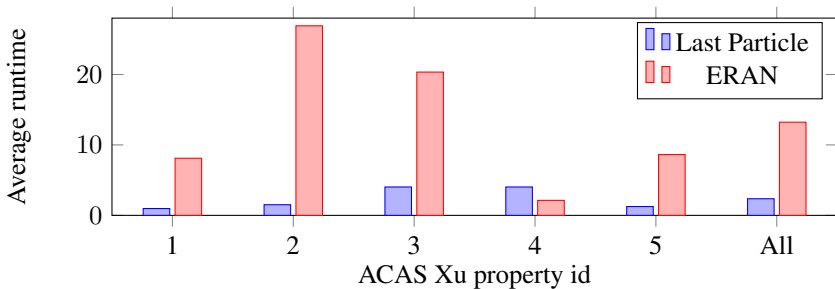

Figure 2: ACAS Xu – runtimes in sec. of ERAN (Deep Zonotope) and Last Particle algorithm $[N = 2, p_c = 10^{-50}, t = 40]$

**Acknowledgement and Disclosure of Funding** We thank Arnaud Guyader for fruitful discussions. This work is funded by ANR/AID Chaire SAIDA. KT gratefully acknowledges PhD funding via a grant from the AID (Agence pour l'Innovation de Défense). In addition we thank the reviewers for their detailed and relevant comments.

## Broader Impact

As deep learning applications move to the physical world it is crucial to understand to what extent and in which situations their predictions can be trusted. Both adversarial and corruption robustness assessment methods are a key step towards building trustworthy deep-learning based cyber-physical systems. At the same time limits of such methods should always be clearly established and their dependence on data explicit.

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
