# A   Appendix: Approximations for the computation of $m$

Providing a very low critical probability $p_c$ means that certification occurs when the simulation ends after a large number of iterations $m$. $\Lambda(L_m)$ follows a Gamma distribution $\Gamma(m, N)$ which can be then approximated by the Gaussian law $\mathcal{N}(m/N; m/N^2)$ (application of the Central Limit Theorem). We introduce $\ell_c$ the threshold associated to $p_c$ s.t. $p_c = \mathbb{P}(h(\mathbf{X}) > \ell_c)$, and $m_c = \log(p_c)/\log(1 - 1/N)$.

Under this assumption:

$$\mathbb{P}(\Lambda(L_m) < \Lambda(\ell_c)) = \alpha \rightarrow \Lambda(\ell_c) = \frac{m}{N} - z_\alpha \frac{\sqrt{m}}{N} \tag{15}$$

with $z_\alpha = \Phi^{-1}(1 - \alpha) > 0$ for $\alpha < 1/2$ and $\Lambda(\ell_c) = -\log(p_c)$. We find a first approximation of $m$ by solving this second order polynomial in $\sqrt{m}$:

$$m \approx \tilde{m}_1 = \left\lceil \frac{1}{4} \left( z_\alpha + \sqrt{z_\alpha^2 - 4N \log(p_c)} \right)^2 \right\rceil. \tag{16}$$

This clearly shows that the dependence on $p_c$ is approximately logarithmic. Table 5 shows that this approximation is excellent even for large $p_c$.

Moreover, if $N$ is large enough, then $N \log(p_c) = N m_c \log(1 - 1/N) \approx m_c$ and $m$ approximately satisfies

$$m - z_\alpha \sqrt{m} - m_c = 0, \tag{17}$$

producing

$$m \approx \tilde{m}_2 = \left\lceil \frac{1}{4} \left( z_\alpha + \sqrt{z_\alpha^2 + 4m_c} \right)^2 \right\rceil = \left\lceil m_c \left( \sqrt{1 + z_\alpha^2/4m_c} + \frac{z_\alpha}{2\sqrt{m_c}} \right)^2 \right\rceil. \tag{18}$$

This shows that $m$ is a little larger than $m_c = \log(p_c)/\log(1-1/N)$.

# B   Experiments in the idealized case

This appendix details the experimental results of Sect. 5.1. This section assumes that $\mathbf{X} = \mathbf{x}_o + \sigma \tilde{\mathbf{X}}$ with $\tilde{\mathbf{X}} \sim \mathcal{N}(\mathbf{0}_n; \mathbf{I}_n)$ and that $h(\mathbf{x}) = \mathbf{x}^\top \mathbf{g} - \tau$ with $\mathbf{g} \in \mathbb{R}^n$ and $\|\mathbf{g}\| = 1$ (w.l.o.g.). In this textbook case, the true probability $p = \pi_0(h(X) > 0)$ depends on $\tau$ by

$$p = 1 - \Phi\left( \frac{\tau - \mathbf{x}_o^\top \mathbf{g}}{\sigma} \right). \tag{19}$$

We now explain how to 'directly' sample a new particle as required by line 11, Alg. 1 for this particular case, without resorting to Alg. 2.

The projection of $\tilde{\mathbf{X}}$ onto $\mathbf{g}$ is Gaussian distributed. By linearity of the score function, conditioning on the event $\mathcal{E} := \{h(\mathbf{X}) > L\}$ means that the c.d.f of $Z := \tilde{\mathbf{X}}^\top \mathbf{g}$ equals:

$$F_Z(z) = \mathbb{1}(z > L_0) \cdot \frac{\Phi(z) - \Phi(L_0)}{1 - \Phi(L_0)} \quad \text{with } L_0 := (L - \mathbf{x}_o^\top \mathbf{g})/\sigma. \tag{20}$$

On the other hand, the projection of $\tilde{\mathbf{X}}$ onto any other direction orthogonal to $\mathbf{g}$ remains normal distributed. This justifies the following construction:

$$Z = F_Z^{-1}(U) = \sigma \Phi^{-1}\left( (1 - \Phi(L_0/\sigma))U + \Phi(L_0/\sigma) \right) \quad \text{with } U \sim \mathcal{U}_{[0,1]} \tag{21}$$

$$\mathbf{X} = \mathbf{x}_o + \sigma \left( Z\mathbf{g} + (\mathbf{I}_n - \mathbf{g}\mathbf{g}^\top)\mathbf{N} \right) \quad \text{with } \mathbf{N} \sim \mathcal{N}(\mathbf{0}_n; \mathbf{I}_n), \tag{22}$$

In a nutshell, $(\mathbf{I}_n - \mathbf{g}\mathbf{g}^\top)$ is the projection onto the $(n - 1)$-dimension subspace orthogonal to $\mathbf{g}$. This operator resets the projection of $\mathbf{N}$ onto $\mathbf{g}$, which is then set to $Z$. Section 5.1 uses this toy example to illustrate our procedure in the idealized case.

## C  Choice of $N$ and $T$

Most experiences are run with $N = 2$ which is counter-intuitive. In this section we elaborate on the choice of $N$ and $T$ using experiments in case of linear decision function and $X$ follows a Gaussian law. More precisely we take $X \sim \mathcal{N}(0, I_d)$ and the score function $s : \mathbb{R}^d \ni x \mapsto x^T n$ with $n \in \mathbb{R}^d$ defining the normal vector of the decision hyperplane. For simplicity, we take $n = e_1$ i.e. the first vector of the canonical basis of $\mathbb{R}^d$. With this toy model the probability of failure for a threshold level $L$ is given by,

$$p = \mathbb{P}(s(X) > L) = \mathbb{P}(X_1 > L) = 1 - \Phi(L) \tag{23}$$

We now apply the last particle algorithm 1 to the statistical test with null hypothesis $\mathcal{H}_0 : p \geq p_c$ and alternative hypothesis $\mathcal{H}_1 : p < p_c$. For numerical experiments below, we take $p = p_c$ and $\alpha = 0.05$. We let vary the number of particles $N$ in the range $\{2, 20, 100\}$ and the parameter $T$ in the range $\{25, 50, 100, 150, 200\}$. For each couple of parameters $(N, T)$ we make 1000 runs and count the number of false positive (i.e. the number of times the algorithm wrongfully asserted that $p < p_c$). The results are presented in the table 5 below.

Table 5: Estimation of false positive rates and number of calls in function of $T$ and $N$ for a toy model

| $N$ | $T$ | Estimated false positive rate | Avg. number of calls |
|---|---|---|---|
| 2 | 25 | 0.038 | 1.05e+03 |
| 2 | 50 | 0.041 | 2.08e+03 |
| 2 | 100 | 0.033 | 4.14e+03 |
| 2 | 150 | 0.026 | 6.19e+03 |
| 2 | 200 | 0.040 | 8.28e+03 |
| 20 | 25 | 0.034 | 1.04e+04 |
| 20 | 50 | 0.050 | 2.07e+04 |
| 20 | 100 | 0.048 | 4.15e+04 |
| 20 | 150 | 0.043 | 6.20e+04 |
| 20 | 200 | 0.043 | 8.29e+04 |
| 100 | 25 | 0.036 | 5.19e+04 |
| 100 | 50 | 0.052 | 1.04e+05 |
| 100 | 100 | 0.049 | 2.07e+05 |
| 100 | 150 | 0.033 | 3.11e+05 |
| 100 | 200 | 0.050 | 4.15e+05 |

## D  Automatic control of kernel strength

In practice the strength parameter $s$ of the kernel is adapted at each iteration using an heuristic. More precisely we choose a acceptance ratio threshold $a_* \in [0, 1]$ and at iteration $k$, after the line 11 of Algorithm 1, decrease the $s$ by a decay rate $0 < \gamma < 1$. Conversely if the acceptance ratio is high but progress, as measured by the relative gain between the old and the new level, is too slow we increase $s$ by the same parameter $\gamma$. This tuning mechanism is further outlined in algorithm 3. Experimentally we find that, with well chosen parameters $(a_*, g_*, \gamma)$ this adaptive tuning speeds up the algorithm drastically keeping both acceptance ratio and level-wise progress under control.

## E  Proof of proposition 4

$\pi_0$ denotes the reference probability distribution. The proof applies to the last particle algorithm describes in Alg.1 in the case where the refreshed particle state $\mathsf{Gen}(l, 1)$ is given for each $l \in \mathbb{R}$ by Alg.2. We recall that in Alg.2, $\mathsf{Gen}(l, 1)$ is obtained by $t$ iterations of a proposal $K$ with score-based accept /reject; starting from a uniformly chosen other (surviving) particle with score strictly greater than $l$.

The proof is based on a (instructive and explicit) probabilistic coupling between this last particle algorithm and the 'idealized algorithm' counterpart. The latter is obtained by taking for $\mathsf{Gen}(l, 1)$ the

---
**Algorithm 3** Adaptive Sampling for one particle $\mathsf{AdaptGen}(L, 1)$
---
**Require:** threshold $L$, finite set $\mathcal{X}$ of particles whose score is larger than $L$, input strength parameter $s_{in}$, scaling parameter $\gamma < 1$, acceptance ratio threshold $a_*$, gain threshold $g_*$
**Ensure:** new particle $\mathbf{X}$, new strength parameter $s_{out}$
 1: **Initialize** $\mathsf{Count} \leftarrow 0$, $s_{out} \leftarrow s_{in}$, $\mathbf{X} \leftarrow \mathcal{U}(\mathcal{X})$ ▷ Draw uniformly a particle in $\mathcal{X}$
 2: **for** $k = 1 : t$ **do**
 3:     $\mathbf{Z} \leftarrow K(\mathbf{X}, s_{in})$ ▷ $\pi_0$ reversible proposal. See Sect. 4.1
 4:     **if** $h(\mathbf{Z}) > L$ **then** ▷ Rejection
 5:         $\mathbf{X} \leftarrow \mathbf{Z}$
 6:         $\mathsf{Count} \leftarrow \mathsf{Count} + 1$
 7:     **end if**
 8: **end for**
 9: **if** $\mathsf{Count} < t \times a_*$ **then**
10:     $s_{out} \leftarrow \gamma \times s_{in}$ ▷ Decrease $s$ if acceptation rate is too low
11: **else**
12:     $L_* \leftarrow \min(h(\mathbf{X}), \min_{x \in \mathcal{X}} h(x))$
13:     $\mathsf{Gain} \leftarrow \frac{L_* - L}{|L|}$
14:     **if** $\mathsf{Gain} < g_*$ **then**
15:         $s_{out} \leftarrow \frac{s_{in}}{\gamma}$ ▷ Increase $s$ if the progress is too low
16:     **end if**
17: **end if**
18: **return** $\mathbf{X}$, $s_{out}$
---

exact conditional distribution $\pi_0(d\mathbf{x}|h(\mathbf{x}) > l)$. The underlying idea (see Guyader et al. [2011]) is that the Markov chain generated by $\mathsf{Gen}(l, 1)$ in Alg.2 leaves invariant the distribution $\pi_0(d\mathbf{x}|h(\mathbf{x}) > l)$, so that the idealized algorithm is formally the limit of the simulated algorithm when $t \to +\infty$.

Step 0: Checking the lower bound assumption

The lower bound assumption can be rewritten as follows:

$$\exists\, p_* > 0, s_0 > 0, \forall\, \mathbf{x}, s \geq s_0, \quad \mathrm{Law}(K(\mathbf{x}, s)) \geq p_* \pi_0 \qquad \text{(Doeblin)}$$

where inequality between two measures simply means that their difference is a non-negative measure. (Doeblin) is a well-known irreducibility condition coined 'Doeblin condition' in the probabilistic literature on Markov chain.

Let us check that the lower bound condition is compliant with some very minor variants of the transformation method detailed in Sect. 4.2.

Consider for instance the transformation: $\mathbf{X} \sim \mathcal{U}(\mathcal{B}_{2,\epsilon}(\mathbf{x}_o))$, $T(\mathbf{U}, \mathbf{x}_o) = \mathbf{x}_o + \epsilon \mathbf{U}(1 : n)$ where $\mathbf{U}$ is $n + 2$-dimensional with uniform distribution on the unit sphere of $\mathbb{R}^{n+2}$.

On the other hand, consider the proposal on the unit sphere of $\mathbb{R}^{n+2}$ obtained by composing the Gaussian proposal (14) in $\mathbb{R}^{n+2}$ with an additional orthogonal projection. This proposal on the sphere has the following two properties: i) it is reversible with respect to the uniform distribution on the sphere (by a symmetry argument), ii) its density satisfies (Doeblin) (by lower bounding (14) with initial condition on the unit sphere by a centered Gaussian distribution).

Combining the latter proposal with $T$ we obtain again a proposal reversible w.r.t. $\mathcal{U}(\mathcal{B}_{2,\epsilon}(\mathbf{x}_o))$ and satisfying (Doeblin). See below for possible (slight but technical) generalizations to proposals satisfying weaker versions of (Doeblin).

Step 1: Uniform rejection rate The acceptance rate of a proposal satisfying (Doeblin) with accept rule given by score $h(\mathbf{x}) > l$ is bounded from below by:

$$p_* \mathbb{P}(h(\mathbf{X}) > l),$$

which is, in turn, uniformly bounded from below if $l \leq l_0$ with $\mathbb{P}(h(\mathbf{X}) > l_0) > 0$.

Note that the proof is thus compliant with the tuning of the proposal strength $s$ w.r.t. a constant rejection rate (App. D), since that latter can be carried out while ensuring (Doeblin).

Step 2: Coupling of proposals Let us define the 'local' coupling between proposals that will enable the coupling between algorithms. Let $\mathbf{x}, \mathbf{x}'$ be given, as well as a proposal satisfying (Doeblin). A coupled proposal $K((\mathbf{x}, s), K(\mathbf{x}, s))$ is generated as follows: i) with probability $p_*$, generate a successful coupling $K(\mathbf{x}, s) = K'(\mathbf{x}', s)$ with distribution $\pi_0$; ii) else, generate independent proposals $K(\mathbf{x}, s)$ and $K'(\mathbf{x}', s)$ with respective distributions $\mathrm{Law}(K(\mathbf{x}, s)) - p_*\pi_0$ and $\mathrm{Law}(K(\mathbf{x}', s)) - p_*\pi_0$.

Clearly, the associated two marginal distributions of $K(\mathbf{x}, s)$ and $K'(\mathbf{x}', s)$ are respectively $\mathrm{Law}(K(\mathbf{x}, s))$ and $\mathrm{Law}(K(\mathbf{x}', s))$.

Step 3: Coupling of the two algorithms

Let us denote by $L_k$ and $L'_k$ the two levels of the last particle at iteration $k$ in Alg. 1 for the *real and idealized algorithms, respectively*. If $L_k = L'_k$, we sample independently $\mathbf{X}'_k$, the new, refreshed particle of the idealized algorithm, according to the exact conditional distribution $\pi_0(d\mathbf{x}|h(\mathbf{x}) > L_k)$ (this replaces line 1 in Alg. 2). $\mathbf{X}'_k$ is then modified in parallel with the new particle of the real algorithm according to Alg. 2 by iterating $t$ times the coupled proposal transition of Step 2; $K$ being used for the real and idealized algorithms, respectively.

After $t$ iterations one has thus obtained a successful coupling with probability (conditional on $L_k$)
$$1 - (1 - p_*\mathbb{P}(h(\mathbf{X}) > L_k))^t \xrightarrow[t \to +\infty]{} 1.$$

Moreover, since Alg. 2 leaves invariant the conditional distribution $\pi_0(d\mathbf{x}|h(\mathbf{x}) > L_k)$, it does not modify the distribution of the refreshed particle in the idealized algorithm.

Step 4: Conclusion by induction

Let $l_0$ be any critical level such that $\pi_0(h(\mathbf{X}) > l_0) > 0$. We consider the following induction hypothesis at iteration $k$:

$H_k$ On the event, $L_k \leq l_0$, The probability that the two particle systems are equal tends exponentially fast to 1 when $t \to +\infty$.

Assume $H_k$ is true. The probability that the two particle systems are equal at iteration $k + 1$ is the probability conditioned by equality at iteration $k$ multiplied by probability of equality at iteration $k$. If the score level is below $l_0$, the former conditioned probability is bounded below by $1 - (1 - p_*\mathbb{P}(h(\mathbf{X}) > l_0))^t$ by Step 3 so that using $H_k$ the induction on $H_{k+1}$ is complete.

We deduce that $\mathbb{P}(L_m \leq l_0)$ converges exponentially fast with $t$ large towards $\mathbb{P}(L'_m \leq l_0)$ for each $l_0$. Using in addition Theorem 1 on the idealized algorithm, we conclude the proof.

Possible Generalizations: It is possible to relax the irreducibility condition (Doeblin) so that it is verified by most practical proposals, see Sec. 4.2. This requires using so-called Lyapounov functions, as well as an extra (but mild) assumption on the shape of $h$ 'at infinity'.

For instance, consider the Gaussian proposal (14) in $\mathbb{R}^{n+2}$. It satisfies the Doeblin condition (Doeblin), but only locally, for all $\mathbf{x}$ in a ball, $p_*$ depending now of the size of the ball.

The extra assumption on the shape of the score function $h$ at infinity is then necessary to check that the rejection rate is again uniformly bounded from below.

Finally, one can remark that the following so-called Lyapounov condition $\mathbb{E}[|K(\mathbf{x}, s)|^2] \leq \rho|x|^2 + c$ holds true (with $\rho = \frac{1}{1+s^2} < 1$ and $c = \frac{s^2}{1+s^2} < +\infty$). It ensures that the proposal cannot be stuck at infinity, in areas where the 'local' Doeblin condition is poor.

One can then couple proposals using (Doeblin) as above, but only when the coupled initial states are in a given ball, and use the Lyapounov condition (see Hairer and Mattingly [2011]) to nonetheless obtain a successful coupling with a lower bounded success rate.

The proof then works as above.

Final remarks: Note that the exponential convergence rate obtained in the proof of Proposition 4 is too sub-optimal to be suitable for practical purpose. Practical estimation of this rate is left for future work although estimating the mixing rate of such Markov chain is known to be difficult and widely dependent on the geometry of $h$.

# F  Implementation details of the experiments

In this section we give further details on the implementations used in the experiments. The source code provided can be used to re-run experiments or run different experiments (see the README for more information).

## F.1  ACAS Xu

In the experiments on the ACAS Xu DNN compression case study we used the 45 neural networks from the VNNLIB website (in ONNX format), which do not require normalizing the inputs. We only tested the 5 first properties since they apply to all networks. We use an adaptive procedure to tune the strength parameter $s$ as explained in D. Experiments main parameters are set to: $N = 2, p_c = 10^{-50}, T = 40, \alpha = 10^{-3}$. We initialize the strength $s$ at $1.5$ and use the adaptive sampling procedure of section D with $\gamma = 0.99, a_* = 0.90, g_* = 0.01$. In addition we ran experiments with the ERAN complete certification method using DeepPoly and Mixed Integer programming on the same benchmark.

## F.2  MNIST

We selected 4 neural networks from the ERAN benchmark: 3 architectures of varying complexity trained with pytorch named `'convMedGRELU__PGDK_w_0.1'`,`'ffnnRELU__PGDK_w_0.1_6_500'` & `'ffnnRELU__Point_6_500'` and a simpler model trained with tensorflow `'mnist_relu_9_200'`. We use batched version of the Last Particle algorithm where we test the local robustnes aroung 100 images in parallel. For each image we create a system of $N(= 2)$ particles and we call the score at each iteration (line 6 in Algorithm 1) with a batch consisting of all lower-scored particles. This trick accelerates the computations by taking advantage of the GPU. We also used the adaptive tuning of the strength, initializing $s$ at $1.5$ and with $\gamma = 0.999, a_* = 0.90, g_* = 0.01$.

## F.3  ImageNet

Similarly to MNIST we used a batched version of the Last Particle algorithm presented in section 3. Again we also used a automatic control mechanism (see section D, initializing $s$ at $1$ and taking $\gamma = 0.999, a_* = 0.90, g_* = 0.01$. For ImageNet we could not run the ERAN certification methods unfortunately since these methods barely scale to such high input dimension and management of ImageNet is not implemented for now on the ERAN GitHub repository.