# OpenReview forum: "Efficient Statistical Assessment of Neural Network Corruption Robustness"
_NeurIPS.cc/2021/Conference — NeurIPS 2021 Poster_

### Official Review · Reviewer_Lats · 2021-07-07

**Rating:** 7
**Confidence:** 4

**Summary:**

The paper approaches the problem of the robustness assessment of neural networks. To be more precise, it focuses on the corruption robustness, which considers the modification of inputs with random noise instead of adversarial attacks. The main question here is to estimate the probability of the correct classification for the given network and noise model.

The main development of the paper is the application of the algorithm proposed in (Guyader, 2011) to the robustness estimation. The algorithm iteratively updates a set of particles by simultaneously sampling from the given distribution (noise distribution in this case) and maximizing the target function. For the target function, the authors consider the classification error. Then, using Theorem 1 from (Guyader, 2011) the authors estimate the probability of false classification and certify whether this probability is below the user-specified threshold.


(Guyader, 2011) Guyader, Arnaud, Nicolas Hengartner, and Eric Matzner-Løber. "Simulation and estimation of extreme quantiles and extreme probabilities." Applied Mathematics & Optimization 64, no. 2 (2011): 171-196.

**Limitations And Societal Impact:**

The authors motivate the proposed algorithm by military applications in the introduction. They also provide the empirical study on the "Airborne Collision Avoidance System X for unmanned aircrafts" dataset. I think the Broader Impact section is incomplete without the discussion of these applications.

**Main Review:**

Overall the proposed scheme sounds very reasonable and technically developed. To the best of my knowledge, the scheme is novel, although I might be unfamiliar with some of the relevant works. A surprising outcome of the provided analysis is that a lesser number of particles yields a better estimation of the error probability. This can be explained by the growing variance of the target function estimate, which assigns more probability mass to high values. Since we are interested in the quantiles around high values, the high variance is favorable. However, to perform the analysis the authors approximate Gamma distribution with the Gaussian using the central limit theorem. Since the tails analysis is important for this application I would expect a more careful study or some discussion of this phenomenon at least.

Unfortunately, the empirical study is a weak point of this paper. As I understand the target application, the algorithm should be tested for the certification error and for the running time. The authors provide the certification error for the ACAS Xu dataset in Table 2 but do not provide the running times for this dataset. For MNIST, the proposed algorithm runs faster than the competitor, but falsely certifies some networks as robust, hence, making errors. Thus, no clear conclusion can be made in terms of the overall performance (both running time and accuracy). Finally, for ImageNet, the authors do not compare against other algorithms at all. For both ImageNet and MNIST the authors test their algorithm on 5 images. I wonder why such a small number of images has been considered, especially when the algorithm runs quickly even on a laptop (as claimed by the authors).

I think the lack of systematic comparison against other algorithms (from different approaches outlined in the introduction) is a significant flaw of the paper. This comparison has to be done even if it's obvious that the proposed algorithm has better properties. For instance, I can imagine that the considered task is simple enough that an adversarial attack via gradient descent can find a counter-example whenever it is possible. Such a baseline would hint readers what to use and what to try in their concrete example.

**Time Spent Reviewing:**

4

---

> ### Author Response · Authors · 2021-08-10
> **Response to reviewer Lats**
>
> We thank reviewer Lats for their analysis of our work.
>
> 1. “However, to perform the analysis the authors approximate Gamma distribution with the Gaussian using the central limit theorem. Since the tails analysis is important for this application I would expect a more careful study or some discussion of this phenomenon at least.”
>
> We strongly disagree. First, the true number of iterations is given by a closed form in the paper (through the incomplete gamma function), and true figures are provided in Table 1. Second, this is a tail analysis only if $\alpha$ is small. Note that even for $\alpha=0.01$, the Gaussian approximation is accurate (the difference is less than 2% of the true number of iterations in Table 1). Third, in practice we never use the Gaussian approximation, we use the close form. The Gaussian approximation is just here to show that this number of iterations scales as O(log(1/pc)) for a fixed alpha.
>
> 2. “The authors provide the certification error for the ACAS Xu dataset in Table 2 but do not provide the running times for this dataset.“
>
> This is not true. The ACAS Xu running times for Last particle and ERAN DeepPoly are presented in Figure 2.
>
> 3. “but falsely certifies some networks as robust, hence, making errors.”
>
> The introduction clearly states that our algorithm verifies whether the probability of failure is inferior to critical level p_c. This is not ideal verification. Yet, the literature acknowledges that ideal verification with formal proof does not scale (NP hard problem). In practice, SMT solvers sometimes give up rendering inconclusive results to the user. At least, when our method makes errors, the user knows that the probability of failure is below p_c.
>
> 4. “Finally, for ImageNet, the authors do not compare against other algorithms at all”
>
> As far as we know, no paper on formal methods (even with approximations) proposes results on ImageNet, and the same holds for statistical reliability papers except PROVERO. Similarly Webb et al. do not tackle this dataset.  Yet, PROVERO (Baluta et al. 2020) uses a critical level $p_c = 10^{-3}$ (denoted \theta in their paper), whereas we use $p_c = 10^{-15}$. In critical applications such as aerospace, this is the order of magnitude required in safety standards [1].
>
> [1]: Jérôme Morio, Mathieu Balesdent. Estimation of Rare Event Probabilities in Complex Aerospace and Other Systems: A Practical Approach. Elsevier. Woodhead Publishing, 2015, 978-0-08-100091-5. ⟨10.1016/B978-0-08-100091-5.09981-0⟩. ⟨hal-01299041⟩.
>
> 5. “MNIST the authors test their algorithm on 5 images.”
>
> This is not true. For MNIST, we use 100 images and 4 networks, hence 400 simulations for each value of epsilon.
>
> 6. “I think the lack of systematic comparison against other algorithms (from different approaches outlined in the introduction) is a significant flaw of the paper.”
>
> This comparison is made harder by the fact that implementations of other methods are not available or only partially available. As an example, the method ‘PROVEN’ which applies formal verification to corruption robustness (and thus would have been our benchmark of choice) has no available implementation to our knowledge. Even for ERAN, some basic operations are not supported (e.g. transposition) which limits the choice of neural architectures and thus grounds for comparison. A ‘systematic comparison’ is thus not possible.
>
> 7. “For instance, I can imagine that the considered task is simple enough that an adversarial attack via gradient descent can find a counter-example whenever it is possible.”
>
> There are comparisons with adversarial attacks in the literature, for instance in Baluta et al. But, as far as we know, there is no adversarial attack with such a guarantee: find whether probability of failure is greater than a given threshold.
>
> Either one is interested in certification and formal methods (like ERAN DeepPoly) are sounder than adversarial attacks, or one is interested in statistical reliability and adversarial attacks do not provide any clue about the probability of failure even under a simple model like Gaussian uncertainties.
>
> 8. “They also provide the empirical study on the "Airborne Collision Avoidance System X for unmanned aircrafts" dataset. I think the Broader Impact section is incomplete without the discussion of these applications.”
>
> ACAS Xu is an aircraft collision avoidance system thus it is meant a priori for both the civil society and the military.
> In the Broader Impact section we addressed cyber-physical systems as a whole but we understand that military applications specifically call for caution. Thank you for this comment.

---

> > ### Comment · Reviewer_Lats · 2021-08-25
> > **thank you for the clarifications**
> >
> > Thank you for the detailed feedback!
> >
> > Indeed, I've missed several details, and now some of my concerns are addressed. However, I still think that the empirical study should be done with a larger number of test images on both MNIST and ImageNet. Also, I would propose to add some clarifications on why the method operates better for a smaller number of particles N. It seems counter-intuitive since the set of particles gives some approximation of the space and values of the target function h, and the higher number of particles should yield better approximation. Though I understand the analysis provided in the paper, which yields this result.
> >
> > Nevertheless, I would like to raise my score. If the authors would provide scaled results as they claim in the response to @Aovq, I would raise it more.

---

> > > ### Author Response · Authors · 2021-08-27
> > > **Response to reviewer Lats feedback**
> > >
> > > Thank you for the thorough consideration of our feedback.
> > >
> > > 1. "However, I still think that the empirical study should be done with a larger number of test images on both MNIST and ImageNet."
> > >
> > > We understand your concerns. As pointed out in the last answer to reviewer Aovq, we are not quite sure we can share supplementary results/material during the review process. However if you can confirm this is possible we can send results on 100 images for ImageNet, obtained on a Tesla V100 gpu.
> > > In addition, we would like to highlight that the code provided in supplementary material can be used to run larger experiments on any Ubuntu system with Tensorflow (v2)/Numpy/Scipy and Pandas installed.
> > > We hope the readme file is detailed enough and we will publish a Github repository containing further results and code extensions.
> > >
> > > 2. "I would propose to add some clarifications on why the method operates better for a smaller number of particles N."
> > >
> > > In deed this is counter-intuitive. This is linked to the fact that the sequential Monte Carlo algorithm is here used for statistical testing rather than estimation. We plan to add a small section in the appendix giving further explanations on this choice.

---

> > > > ### Comment · Reviewer_Lats · 2021-08-30
> > > > **response to the authors**
> > > >
> > > > Thank you once again for the comments. I think that sharing additional experimental results is not a problem if they are requested.
> > > >
> > > > Anyway, I believe that the authors will include additional results and will polish the paper. I'm not an expert in the field, hence, I could be missing something. From my perspective, the paper could be indeed interesting for the community, and the authors have adequately addressed my major concerns.

---

### Official Review · Reviewer_Aovq · 2021-07-16

**Rating:** 7
**Confidence:** 3

**Summary:**

The authors provide a novel way of accessing neural network robustness. In the classification by [Singh et al., 2018] they provide a method that is complete, but not sound. In other words, their method is guaranteed to certify the model if it satisfies the property, but is capable of producing False Positives - e.g. models that do not satisfy the property can be certified. The derived method is based on statistical hypothesis testing - rather than looking at worst case scenarios, which is done often, they look at the probability of violation of the property over a density of perturbations. The core part of the method is the Last Particle algorithm, invented by [Guyader et al. [2011]], which come with a theorem that states the distribution of its output. This provides the mathematical framework needed to perform a hypothesis testing given a run of the algorithm. The authors demonstrate with several propositions, proved in the Appendices, how the critical probability p_c and the significant level \alpha affect the power of the test (it basically produces less False Positives as \alpha and p_c decrease). In addition the authors demonstrate that the proposed method scales logarithmically in the number of calls needed, which is in contrast to other approaches in literature that have linear dependence. Several empirical experiments validate the claims and showcase that indeed the method provides a computationally efficient method, with very good results.

**Limitations And Societal Impact:**

As indicated by the authors, the two major limitations are the fact that the method is not parallelizable and does not make any use of model information (in the form of gradients or anything else). I think I would have liked them to provide some more detailed discussion on how this can be extended in the future, or if why there are fundamental blocks to it. For instance one can run multiple chains of the proposed algorithm in parallel, which would provide multiple draws from the same distribution and potentially these could be combined to improve the confidence level bound (though this is speculation, have not actually tried to prove this). In addition, the proposal distributions for resampling the new point seem relatively simple. Since they are looking for a MC that has as invariant distribution \pi_0, I wonder why they have not even discussed any more sophisticated MCMC methods (or why those would not be applicable to this problem).

Finally, the ImageNet experiments seem a bit small - only 5 test images? Given that this runs on a laptop for 30sec per image, you could have run this over 1500 images per day (and this assuming you only have a single laptop!). I would like to see here a significantly broader evaluation, and with access to even several computers you can probably run this on the whole ImageNet test set. Also there is no comment on running ERAN on that example - how expensive is it or does it always reach infeasible results? From the results on MNIST it seems that it is an order of magnitude slower, but if that implies 30 minutes per image, this is still something that you can definitely run and show us the results.

**Main Review:**

The paper is written really good and is easy to follow with clear narrative, proposed method and experimental results.
Unfortunately, I'm not too familiar with this area of research, hence I can not comment on how well do the authors present or omit any other competitive approaches.
I think the comparison with ERAN on the simpler problems indeed show very significant promise for this method to be practically applicable. In fact potential combination with the sound solvers, provides both an upper and lower bound on the number of certifications, which can be useful in practice.

**Time Spent Reviewing:**

2

---

> ### Author Response · Authors · 2021-08-10
> **Response to reviewer Aovq**
>
>
> We thank reviewer Aovq for the very constructive comments and suggestions.
>
> 1. “I can not comment on how well the authors present or omit any other competitive approaches.“
>
> The main other competitive approaches for statistical assessment are the paper by Webb et al. (ICLR 2019 - with an `overkilled’ Adaptive Multi-Level Splitting algorithm but no confidence bound) and more recently the paper by Baluta et al (IEEE ICSE 2020 - with an inefficient crude Monte Carlo).
>
>  2. “In fact potential combination with the sound solvers, provides both an upper and lower bound on the number of certifications, which can be useful in practice.“
>
> Indeed this could be a very good application of our algorithm, although it would only be possible when sound verifiers can effectively work.
>
> 3. “the fact that the method is not parallelizable and does not make any use of model information (in the form of gradients or anything else)”
>
> We do provide code with batch parallelization (i.e. parallelization over different inputs) however computations over one input are indeed sequential by design.
> Our method is indeed a ‘black box’ method that does not make use of any model information. Integration of gradient information into a more sophisticated Langevin Sequential Monte Carlo algorithm is our next step. By doing so we hope to get faster methods but we will lose generality.
> As it is for the moment, our method is black box and thus more general:
> In principle, it also works for ‘classical’ machine learning algorithms (i.e. Random Forests, Gradient Boosting Trees), as long as a continuous score (i.e. probability, logits…) can be outputted by the model. Our method works for any network architecture whereas formal methods based on abstract domains need hard-coded translation of each neural network layer/operation (e.g. transposition). For instance,  LSTMs are not yet supported by ERAN.
>
> 4. “For instance one can run multiple chains of the proposed algorithm in parallel ...”
>
> Exactly! This is a good idea, which certainly has a connection with the thesis of Clement Walter..
>
> We think that parallelism is not a drawback since we are usually interested in assessing point-wise reliability over a set of data points. We prefer to run several simulations (ie. from different starting data points x_o) in parallel.
>
> 5. “the proposal distributions for resampling the new point seem relatively simple…. I wonder why they have not even discussed any more sophisticated MCMC methods (or why those would not be applicable to this problem).”
>
> That’s partially correct. As compared to the blunt proposal used in Webb and al., we have used a Gaussian reversible proposal in conjunction with transformations (a classical tool in Statistical Engineering Reliability and Rare Event Simulation). We believe it constitutes a more efficient sampling (shorter burn-in period, hence less network calls). Other MCMC methods, in particular those using gradient information,  (like Langevin/Hamiltonian MCMC) are tempting options, but require non-singular target distributions, hence a modification of the setting. This is our next step.
>
> 6. “Finally, the ImageNet experiments seem a bit small.”
>
> We agree. Our bet was to demonstrate that certification is tractable on a laptop, even on ImageNet. In addition, since this is a statistical method, we have to run it several times in order to compute standard deviations on running times. We now have results on more images thanks to a GPU cluster.

---

> > ### Comment · Reviewer_Aovq · 2021-08-24
> > **Thanks for the response**
> >
> > Given that NuerIPS does not allow editing of the manuscript, could you share the results you have on larger part of ImageNet.
> >
> > Would it also be feasible to include some of the discussion in your answer, regarding extensions and limitations of the MC method in the main paper?

---

> > > ### Author Response · Authors · 2021-08-27
> > > **Thanks again for your interesting feedback**
> > >
> > > Thank you again for your useful feedback.
> > >
> > > 1. "Given that NeurIPS does not allow editing of the manuscript, could you share the results you have on larger part of ImageNet."
> > >
> > > As it is we are not sure whether we are allowed to share supplementary material/results during the review process. If you can confirm us this is permitted we will gladly send you simulations (with same parameters as the paper) for 100 images from ImageNet.
> > > We would like to highlight the fact that with the code originally provided in the supplementary material you can rerun experiments or run new experiments with up to 100 images on ImageNet. The code has been tested on Ubuntu system with TensorFlow v2.
> > >
> > > Extended version of this code (with high performance computing using Numba) will be published in a Github repository which is private for now.
> > >
> > > 2. "Would it also be feasible to include some of the discussion in your answer, regarding extensions and limitations of the MC method in the main paper?"
> > >
> > > This is exactly our intention. As you pointed out, more sophisticated MCMC method using gradient information can be used and this is our current research direction. We plan to include this future direction in the conclusion.

---

> > > > ### Comment · Reviewer_Aovq · 2021-08-31
> > > > **Reply**
> > > >
> > > > Thanks for the reply. I'm also not 100% sure on this and might need to ask an AC of whether you can provide further results (and they stay confidential). Nevertheless, I have seen this done in other submissions, but I'll leave it to your discretion.

---

> > > > > ### Author Response · Authors · 2021-09-03
> > > > > **Response to reviewer Aovq**
> > > > >
> > > > > Thank you again for your comments.
> > > > >
> > > > > From what we understood it is in deed possible as long as anonymity is respected.
> > > > >
> > > > > Below you can find the results for 100 images from the ImageNet validation dataset, ran on a Nvidia Tesla  V100 GPU,
> > > > >
> > > > > Network,  Epsilon,  Verified (%),  Calls,  Avg. Compute time (s)
> > > > >
> > > > > MobileNet,  0.02,  71,  1385,  27.05
> > > > >
> > > > > MobileNet,  0.03,  63,  1279,  24.90
> > > > >
> > > > > MobileNet,  0.06,  48,  1030,  19.63
> > > > >
> > > > > ResNet50,  0.02,  80,  1527,  32.99
> > > > >
> > > > > ResNet50,  0.03,  73,  1441,  31.23
> > > > >
> > > > > ResNet50,  0.06,  55,  1171,  25.25

---

### Official Review · Reviewer_K7DT · 2021-08-02

**Rating:** 5
**Confidence:** 3

**Summary:**

The paper studies (instance-wise) certification for Deep Neural Networks. Quantitatively, this metric can be defined as an integral, and the common computational approach is using Monte Carlo simulation.

The authors consider using statistical hypothesis testing as a surrogate of certification, and the key contribution is a more efficient sequential Monte Carlo algorithm. The theoretical guarantees of the proposed algorithm are by bounding the False Positive Rate and False Negative Rate. The theoretical results (Prop 1 to Prop 4) rely on the property of the Last Particle simulation, which is presented in Theorem 1.

The authors also demonstrate the efficiency and scalability of their algorithm.

**Limitations And Societal Impact:**

Yes, the authors mentioned that the Last Particle simulation is sequential, which is not GPU friendly.

**Main Review:**

The workhorse of the paper is a more efficient sequential Monte Carlo algorithm by using the last particle simulation. Both the algorithm and the theoretical analysis strongly rely on previous work. The algorithm improves the crude Monte Carlo simulation by using the Last Particle simulation. In the context of certification, the theoretical results are derived by simply bounding the False Positive Rate and False Negative Rate. It seems that the contribution of this work is incremental compared to previously known results.

**Time Spent Reviewing:**

10

---

> ### Author Response · Authors · 2021-08-10
> **Response to Reviewer K7DT**
>
> We thank reviewer K7DT.
>
> 1. “Both the algorithm and the theoretical analysis strongly rely on previous work.“
>
> This is essentially true, like Baluta et al. (IEEE ICSE 2020) rely on crude Monte Carlo, like Webb et al. (ICLR 2019) rely on AMLS. We do not claim to invent a new Monte Carlo scheme.
>
> The novelty is the adptation of the Last Particle algorithm for hypothesis testing whereas it was originally used for probability estimation. This yields a tuning of this algorithm to get statistical guarantees (which were absent of Webb. et al.). Another novelty is the scalability that allows reliability assessment of large networks (tackling ImageNet) at a low critical level $p_c = 10^{-15}$ (whereas Baluta et al. used $p_c=10^{-3}$).

---

### Decision · Program_Chairs · 2021-09-28

**Decision:**

Accept (Poster)

**Comment:**

This paper uses a hypothesis testing based approach to provide a statistical assessment of corruption robustness, with a scalable MC-based computational approach.

The authors have made an impressive effort to address the concerns of the reviewers. Though not all the concerns are fully addressed, the emerging consensus is that the additional comments, clarifications, and also experiments, have satisfactorily addressed many of the concerns raised.

**Consistency Experiment:**

NeurIPS has a long history of experimentation. In 2014, NeurIPS ran an experiment in which 10% of submissions were reviewed by two independent committees to quantify the randomness in the review process. This year, we repeated a variant of this experiment to see how the quality of the review process has changed over time.  This paper was part of the experiment and was therefore assigned to two committees (consisting of reviewers, an Area Chair, and a Senior Area Chair) that reached independent decisions.  If both committees made the same recommendation, this recommendation was followed. If a single committee recommended acceptance, the paper was accepted (with the exception of a few cases in which the other committee identified what we considered a fatal flaw, e.g., an error in a key result).

This copy’s committee reached the following decision: **Accept (Poster)**

The other committee assigned to the paper recommended **Reject**.  You can find the other set of reviews, along with any follow up discussion with the authors here:
https://openreview.net/forum?id=RR16clcsH7